# Assessment of efficacy of mutagenesis of gamma-irradiation in plant height and days to maturity through expression analysis in rice

Andrew-Peter-Leon M. T.[1], S. Ramchander[2], Kumar K. K.[3], Mehanathan Muthamilarasan[4], M. Arumugam Pillai[1]*

1 Department of Plant Breeding and Genetics, Agricultural College and Research Institute, Tamil Nadu Agricultural University, Killikulam, Tuticorin, Tamil Nadu, India, 2 Visiting Scientist (SERB–National Post-Doctoral Fellow), IRRI-South Asia Hub, ICRISAT, Patancheru, Hyderabad, India, 3 Centre for Plant Molecular Biology and Biotechnology, Tamil Nadu Agricultural University, Coimbatore, Tamil Nadu, India, 4 Department of Plant Sciences, School of Life Sciences, University of Hyderabad, Telangana, India

* mapillai1@hotmail.com

**Data Availability Statement:** All relevant data are within the manuscript and its Supporting Information files.

## Abstract

Introduction of semi-dwarfism and early maturity in rice cultivars is important to achieve improved plant architecture, lodging resistance and high yield. Gamma rays induced mutations are routinely used to achieve these traits. We report the development of a semi-dwarf, early maturing and high-yielding mutant of rice cultivar 'Improved White Ponni', a popular cosmopolitan variety in south India preferred for its superior grain quality traits. Through gamma rays induced mutagenesis, several mutants were developed and subjected to selection up to six generations ($M_6$) until the superior mutants were stabilized. In the $M_6$ generation, significant reduction in days to flowering (up to 11.81% reduction) and plant height (up to 40% reduction) combined with an increase in single plant yield (up to 45.73% increase) was observed in the mutant population. The cooking quality traits *viz.*, linear elongation ratio, breadthwise expansion ratio, gel consistency and gelatinization temperature of the mutants were similar to the parent variety Improved White Ponni. The genetic characterization with SSR markers showed variability between the semi-dwarf-early mutants and the Improved White Ponni. Gibberellin responsiveness study and quantitative real-time PCR showed a faulty gibberellin pathway and epistatic control between the genes such as *OsKOL4* and *OsBRD2* causing semi-dwarfism in a mutant. These mutants have potential as new rice varieties and can be used as new sources of semi-dwarfism and earliness for improving high grain quality rice varieties.

## Introduction

Rice is the staple food for almost 50% of the world's population. Overcoming the threats caused by biotic and abiotic factors have been an important task in rice breeding. Recently, the loss of arable lands and changing climatic patterns has further increased the pressure to develop cultivars with improved plant architecture, high yield and superior grain quality.

**Funding:** The author(s) received no specific funding for this work.

**Competing interests:** The authors have declared that no competing interests exist

Mutagenesis as a tool can be effectively utilized to improve and modify the genotypes of popular rice cultivars appropriate for the modern agricultural and commercial needs. Improved White Ponni (IWP) is one such rice variety with fine-slender grain, high yield potential, moderate resistance to tungro, rice blast, bacterial blight, mite and green leafhopper. The variety even responds well under organic cultivation systems. However, the tall stature and late maturity of this variety relates to severe lodging and yield losses [1,2].

Continual improvements and studies have shown that semi-dwarfism in rice, conferred by the *sd-1* gene, improves lodging resistance and yield [3]. After the release of IR8 –the miracle rice by IRRI [4], most of the modern rice varieties were developed with the semi-dwarf gene, *sd1*. This accelerated the loss of valuable genetic base which threatens further improvements in rice cultivars. Although new sources of semi-dwarfism in rice plants have been reported before [5–8], the negative effects caused by these genes such as severe dwarfism, reduced panicle length, poor grain yield and poor grain quality have limited their use in rice breeding programs.

In crop plants, mutation breeding has been used as a tool to develop plants with improved architecture such as semi-dwarfism and early maturity together with improved quality traits [9]. Among the different types of mutagens used, the ionizing radiations (physical mutagens) have been widely used.

In rice, mutation breeding has been mainly used to develop semi-dwarfism and earliness [10]. Such cultivars were either directly released as new varieties or used as breeding stocks. In Japan, rice variety *Reimei* (a gamma-ray mutant) was one of the first allele sources used for the development of dwarf rice cultivars [11]. The allele conferring semi-dwarfism in this cultivar was later found to be *sd1* [12]. Gamma-rays was utilized to develop semi-dwarf mutants of cultivars such as Basmati 370 [13,14]. Dominant type of semi dwarf cultivars were also developed through induced mutagenesis: *Ssi1* allele through X-ray irradiation [15]; *Sdt97* allele in a rice mutant [16,17]. Similarly, T-DNA insertion [18], RNA interference [19] and recently CRISPR/Cas9 [20] induced mutations were used to develop semi-dwarf cultivars in rice.

Intercalary meristem cell division and elongation are the major causes for internodal elongation in rice. In dwarf mutants, poor internodal elongation is often associated with defective gibberellin pathway that reduces cell division [21]. Such mutants when supplied with gibberellic acid, would attain rapid internodal elongation similar to the wild types [8]. Hence, it is essential to analyze the gibberellin sensitivity in rice mutants which could relate to the defective gibberellin pathway.

The introduction of semi-dwarfism and early flowering in rice variety Improved White Ponni can improve the lodging resistance. For this objective, mutations were induced in cultivar Improved White Ponni through gamma-irradiation [22]. In this study, we evaluated twenty mutants in advanced homozygous generation (sixth mutant generation: $M_6$) for yield and grain quality traits. The genotypes were tested using SSR markers and quantitative real-time polymerase chain reaction (qRT-PCR).

## Materials and methods

### Development and selection of mutants

In 2011, seeds of Improved White Ponni were treated with different doses of gamma irradiation (100, 200, 300, 400 and 500 Gy) in the Gamma Chamber facility (Model GC1200, Tamil Nadu Agricultural University, Coimbatore, India). The experimental plots were maintained at Agricultural College and Research Institute (Killikulam) and Agricultural Research Station (Thirupathisaram), representing the rice growing tracts of South Tamil Nadu, India. Plant to progeny method was followed to forward individual plants from $M_1$ to $M_2$ [22]. Plants with

semi-dwarfism and earliness were primarily selected and forwarded to $M_3$ [23]. In $M_4$, 159 mutant families were evaluated and 70 were forwarded to $M_5$. From this 70 in $M_5$, 20 mutant families were selected and forwarded to the $M_6$ generation. During 2016, the 20 $M_6$ mutants were evaluated in randomized block design with two replications. The parent variety IWP was grown as the control. For morphological observations, ten plants per treatment per replication were chosen randomly and recorded.

## Morphological observations

Plant morphological traits *viz.*, plant height in cm, days to 50% flowering, number of productive tillers per plant, panicle length in cm, number of filled grains per panicle, thousand grain weight in grams (g) and single plant yield in grams (g) were recorded. Rice grain quality traits *viz.*,dehusked kernel (brown rice) length in mm, dehusked kernel breadth in mm, kernel length to the breadth (L/B) ratio, rice length after cooking in mm, rice breadth after cooking in mm were measured. The linear elongation ratio and breadth-wise expansion ratio were calculated according to standard methods [24].

## Amylose content

The amylose content of the mutant lines and IWP were estimated by colorimetric method [25]. Based on the per cent amylose content the genotypes were categorized (S1 Table).

## Gel consistency

The way cooked rice hardens upon cooling was measured by gel consistency according to the Standard Evaluation System [26].

## Statistical analysis

**Estimation of variance parameters.** The mean, variance and standard error were estimated by following the standard methods [27]. The variances (phenotypic and genotypic) and broad sense heritability were estimated by following the standard methods [28]. The phenotypic and genotypic coefficients of variability calculated by following the standard methods [29]. The genetic advance (as per cent of mean) was calculated according to [30].

**Genotypic correlation.** The genotypic correlation (Pearson correlation coefficients) between the traits was computed using Multi-Environment Trial Analysis with R for Windows (META-R) [31]. The correlation values were plotted using R package 'corrplot' [32].

**Cluster analysis and principal components analysis.** The hierarchical cluster analysis of genotypes based on squared-Euclidean distances and the principal components analysis were performed using R software environment for statistical computing, version 3.5.1 [33]. The PCA biplot drawn using the first two principal components (PC1 and PC2) was overlaid with the hierarchical clusters.

## Molecular analysis with SSR markers

To assess the mutation rate, the mutants were analysed with SSR markers. An SSR marker panel consisting of 53 markers were chosen based on reports of QTL associations with plant height and days to flowering which spread throughout the genome of rice (S2 Table). Genomic DNA of IWP and mutants were isolated from young leaves following a modified cetyl trimethylammoniumbromide (CTAB) method [34]. The polymerase chain reaction was performed with Prime*Taq* 2X mastermix (GCC biotech, India) according to the manufacturer's instruction and the amplicons were electrophoresed in 2% agarose gel and visually scored by

comparing with a standard 100 base pair ladder. The molecular diversity was analyzed using molecular dissimilarity analysis software DARwin (Dissimilarity Analysis and Representation for Windows) version 5.0 [35].

## Scanning electron microscopy of mutants

The internal cell structure of IWP and a semi-dwarf mutant was studied using a scanning electron microscope (SEM) facility (FEL quanta 200 SEM, ThermoFisher Scientific, US) available at Tamil Nadu Agricultural University, Coimbatore. Transverse sections of leaf, nodal region and internodal regions were studied.

## Responsiveness to external GA$_3$

A superior mutant from the M$_6$ generation, designated as WP-22-2 was selected based on the morphological observations. Dwarf mutants in rice are classified as gibberellin responsive on non-responsive based on their phenotypic response to the external application of gibberellin hormone. Gibberellin responsiveness of WP-22-2 was studied by spraying 50 μM gibberellin on 10 days old seedlings (GA$_3$ solution prepared with gibberellic acid crystals, SRL, Mumbai, India). Five days after treatment, 1$^{st}$ internode length and 2$^{nd}$ leaf length of the seedlings were measured and compared with the parent IWP. Mean lengths were compared by using Student's t-test and plotted using R [33].

## Mutation characterization through quantitative real time-polymerase chain reaction

Molecular level changes during external application of GA$_3$ was studied using qRT-PCR. Relative expression levels of six plant height controlling genes (Table 1) was compared at different time-points.

**GA$_3$ treatment.** Fourteen days old seedlings of IWP and WP-22-2 were sprayed with 50 μM GA$_3$ solution using a hand sprayer. Leaf samples were collected and flash-frozen in liquid nitrogen at 0 hrs (control), 6 hrs, 12 hrs and 24 hrs after spraying and stored at -80˚C until RNA isolation.

**RNA isolation and cDNA synthesis.** Total RNA was isolated from the tissue by following the TRIzol RNA isolation protocol [36]. The quality and quantity of the isolated RNA were

**Table 1. Target genes and primers used for qRT PCR.**

| S. No. | Primer ID | Primer sequences (5' → 3') | | Targeted gene | Functions |
|---|---|---|---|---|---|
| 1. | SLR-1 | Forward | CGATCGGGCTTACGGTTCTC | SLR-1 (LOC_Os03g49990) | Probable repressor of GA signalling pathway. Overexpression induces dwarf phenotype |
| | | Reverse | AGATGGGCTAGGAGGACCAA | | |
| 2. | GA | Forward | CCAATTTTGGACCCTACCGC | GA20oxidase (LOC_Os01g66100) | Key enzyme in biosynthesis of gibberellin. Promotes internode elongation |
| | | Reverse | TCCATTCATCCGTCGTTCCA | | |
| 3. | OsKOL4 | Forward | CAGATGACCAACTGATGCTGC | ent-Kaurene oxidase like-4 (LOC_Os06g37300) | Heme binding; gibberellin biosynthetic process |
| | | Reverse | CGGATCTCTTGGTAGAGTAGC | | |
| 4. | KO2 | Forward | AACCTGTACGGGTGCAACAT | ent-kaurene oxidase 2-like (LOC_Os06g37364) | Heme binding; key role in biosynthesis of GA |
| | | Reverse | CTTGTACATGTCCGCCACCT | | |
| 5. | MAX2 | Forward | GACAAATGGGATGGCGTGTG | Fbox/LRR-repeat MAX2 homolog (LOC_Os06g06050) | Mutations cause high tillering and dwarfism |
| | | Reverse | TCAGATTAAATCCTTACTGCTGTGT | | |
| 6. | OsBRD2 | Forward | AAGACATGCTGGTTCCCTTGT | Brassinosteroid Deficient (LOC_Os10g24780) | controls grain shape and height; cell elongation |
| | | Reverse | TGGTTTTCACAGGGAGCTTGT | | |

estimated using 1.2% agarose gel and NanoDrop (Thermo Fisher Scientific, US) spectropho-tometer. The complementary DNA (cDNA) was synthesized using Verso cDNA synthesis kit (Thermo Fisher Scientific, US) with random hexamer and Oligo dT (in 3:1 ratio) as RNA primers in a thermal cycler (ProFlex PCR system, ThermoFisher Scientific, US) following manufacturer's instructions. For template in qRT PCR, cDNA was diluted ten folds with molecular grade water.

**Quantitative Real Time PCR.** PCR was performed with template cDNA and master mix (PowerUp SYBR Green master mix, Thermo Fisher Scientific, US) in Real Time PCR machine (ABi 7900 HT, US) with standard operating conditions. *OsAct* was used as an internal control to normalize the data. The expression ratio of each gene was calculated relative to its expres-sion in control sample by the $\Delta\Delta C_T$ method [37]. Error bars representing standard error were calculated based on three technical replicates for each biological duplicate.

### Gene sequencing analysis

Whole-genome assembly was initiated in IWP and WP-22-2 genotypes (Illumina HiSeq 2500; Agrigenome labs, Hyderabad). Preliminary sequence analysis showed mutations in the *OsGA20Ox2* gene of WP-22-2 mutant. This was confirmed by targeted sequencing of the gene using primers SD1_F [5'-TCCCTCATCCCCTGTGGTG-3'] SD1_R [5'-ATGGCGGGTAGTAGT TGCAC-3'].

## Results

### Mean performance

Plant height and days to 50% flowering was generally reduced in the $M_6$ generation mutants (Table 2; Fig 1). Up to 13 days reduction in fifty per cent flowering (11.8% reduction from 110 days in IWP-control) was observed in a mutant WP 5–4. Up to 11 days reduction in days to flowering was observed in six mutants. The phenotypic co-efficient of variation (PCV) and genotypic co-efficient of variation (GCV) were low (7.23 and 7.17, respectively). The trait recorded high heritability (98.4%) and intermediate genetic advance as per cent of mean (14.7%). Similarly, up to 42% reduction in plant height was observed in WP-16-5 (149.9 cm in IWP-control). High PCV (20.03%) and GCV (20.02%) were observed for the trait. High herita-bility (99.92%) and high genetic advance as per cent of mean (41.23%) was observed.

Increase in yield was observed in many semi-dwarf and early mutants when compared to the IWP-control. Four mutants have recorded single plant yield above 50 grams (Table 2).

An average of 5.0% increase in milling per cent and 4.9% increase in head rice recovery was observed among the mutants while IWP-control recorded the lowest milling per cent of 61.7%.

Reduction in dehusked kernel length (brown rice) was observed in mutants (0.25 mm to 0.6 mm reduction) compared to the IWP-control (5.45 mm). But the dehusked kernel breadth remained unaltered in the mutants.

The linear elongation ratio of rice kernels (LER) in nine mutants were higher than the IWP-control. The gelatinization temperature of the mutant and IWP-control was uniform. Similarly, the gel consistency values, which measure the softness of rice after cooking, were unaltered in the mutants. All genotypes, including IWP-control, had a value of more than 60 mm, corresponding to soft rice grains.

IWP-control and mutants WP 6–3, WP 23–3 and WP 30–1 had low amylose contents. Three mutants, WP 5–4, WP 15–5 and WP 16–5 had intermediate amylose content (S3 Table).

Based on the overall morphological performance, mutant WP-22-2 was selected with high single plant yield, semi-dwarfism, reduced days to maturity than IWP, increased milling per

**Table 2. Mean performance of the M₆ mutants and IWP-control.**

| S. No. | Lines | DFF | PH (cm) | NOPT | PL (cm) | GPP | TGW (g) | SPY (g) |
|---|---|---|---|---|---|---|---|---|
| 1 | WP 5–1 | 119.0 | 127.1 | 19.5 | 24.4 | 264.3 | 15.2 | 47.0 |
| 2 | WP 5–4 | 97.0 | 94.1 | 19.4 | 23.8 | 227.3 | 15.2 | 40.6 |
| 3 | WP 6–3 | 104.0 | 97.1 | 24.2 | 25.1 | 218.3 | 14.3 | 43.3 |
| 4 | WP 6–4 | 102.5 | 90.5 | 16.5 | 24.2 | 235.3 | 14.5 | 32.7 |
| 5 | WP 6–5 | 104.5 | 91.6 | 16.9 | 22.6 | 264.3 | 14.7 | 37.6 |
| 6 | WP 15–1 | 102.5 | 90.5 | 22.3 | 22.7 | 192.8 | 13.3 | 49.4 |
| 7 | WP 15–5 | 102.0 | 98.1 | 23.9 | 22.6 | 198.3 | 14.1 | 59.4 |
| 8 | WP 16–1 | 98.5 | 95.9 | 20.6 | 23.7 | 258.5 | 13.5 | 52.9 |
| 9 | WP 16–2 | 98.5 | 89.3 | 19.7 | 23.6 | 234.8 | 13.7 | 56.0 |
| 10 | WP 16–3 | 107.0 | 90.0 | 18.0 | 23.0 | 257.0 | 13.7 | 31.3 |
| 11 | WP 16–4 | 108.0 | 92.7 | 19.3 | 24.1 | 278.8 | 14.0 | 49.8 |
| 12 | WP 16–5 | 109.5 | 86.5 | 23.1 | 24.3 | 286.0 | 13.7 | 48.2 |
| 13 | WP 22–1 | 104.5 | 88.3 | 20.5 | 22.7 | 269.3 | 13.0 | 38.2 |
| 14 | WP 22–2 | 99.0 | 91.6 | 21.3 | 24.2 | 264.3 | 12.5 | 54.6 |
| 15 | WP 22–3 | 106.5 | 94.2 | 24.4 | 23.8 | 266.8 | 12.8 | 49.7 |
| 16 | WP 22–5 | 99.0 | 89.6 | 21.5 | 24.0 | 242.8 | 12.6 | 47.5 |
| 17 | WP 23–3 | 98.5 | 89.6 | 22.0 | 24.0 | 240.3 | 13.3 | 48.3 |
| 18 | WP 23–4 | 99.0 | 93.6 | 22.0 | 23.3 | 284.5 | 14.1 | 42.9 |
| 19 | WP 30–1 | 122.0 | 149.9 | 22.9 | 24.4 | 191.5 | 16.0 | 35.3 |
| 20 | WP 30–5 | 122.0 | 135.7 | 19.7 | 23.9 | 184.8 | 15.2 | 39.3 |
| 21 | IWP Cont | 110.0 | 149.9 | 21.7 | 26.3 | 224.5 | 15.6 | 40.8 |
| | **Grand mean** | **105.4** | **101.2** | **20.9** | **23.8** | **242.1** | **14.0** | **45.0** |
| | **Range** | 97.0 to 122.0 | 86.5 to 149.9 | 16.5 to 24.4 | 22.6 to 26.3 | 184.8 to 286.0 | 12.5 to 16.0 | 31.3 to 59.4 |
| | **PCV (%)** | 7.23 | 20.03 | 10.80 | 3.81 | 13.07 | 7.17 | 17.17 |
| | **GCV (%)** | 7.17 | 20.02 | 10.70 | 3.68 | 12.91 | 7.12 | 17.12 |
| | **Heritability (%)** | 98.41 | 99.92 | 98.08 | 93.22 | 97.54 | 98.62 | 99.29 |
| | **Genetic advance** | 15.45 | 41.74 | 4.56 | 1.74 | 63.60 | 2.04 | 15.81 |
| | **Genetic advance (as per cent of mean)** | 14.66 | 41.23 | 21.82 | 7.32 | 26.27 | 14.57 | 35.13 |

(DFF- days to 50% flowering; PH- plant height; NOPT- number of productive tillers per plant; PL- panicle length; GPP- grains per panicle; TGW- thousand grain weight; SPY- single plant yield).

cent and head rice recovery, fine grain L/B ratio and high linear elongation ratio (Tables 2 and 3). This mutant was selected for further analyses.

## Correlation between traits

The genotypic correlation between the traits was calculated (S4 Table; Fig 2). High significant positive correlation (Pearson correlation co-efficient: 0.78) was observed between plant height and days to fifty per cent flowering. The trait days to fifty per cent flowering was negatively correlated with number of productive tillers (-0.56). The grain quality trait length after cooking had positive correlation with thousand grain weight (0.86), length before cooking (0.73), panicle length (0.49) and LB ratio (0.45). However, it was negatively correlated with traits head rice recovery (-0.62) and number of grains per panicle (-0.51). The trait single plant yield had no significant correlations with any other traits, however non-significant and negative correlations was observed with days to fifty per cent flowering and plant height.

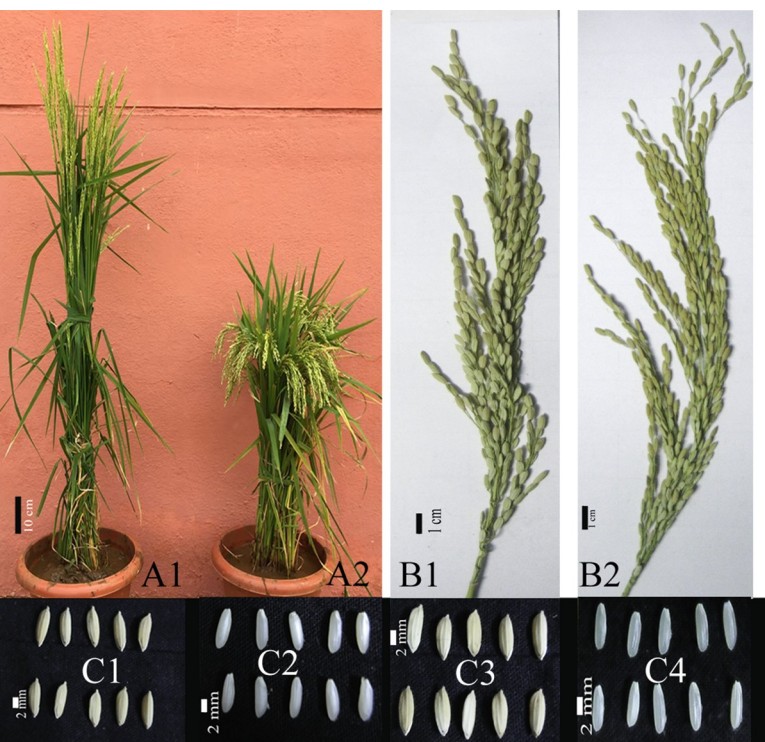

**Fig 1. Comparison of IWP-control and a high yield, semi-dwarf and early maturing mutant.** A1) The parent variety IWP; A2) a semi-dwarf, early maturing and high-yielding mutant, WP-22-2 (note: the IWP is still in the flowering stage while in the WP-22-2, the panicles are already maturing); scale bar– 10 cm; B1) Panicle of the IWP and B2) WP-22-2; scale bars– 1 cm; C1 & C3) Rice kernels of IWP and WP-22-2; C2&C4) dehusked kernels of IWP and WP-22-2; scale bars– 2mm.

### Genetic variability based on hierarchical cluster analysis

Three major clusters were identified in the hierarchical cluster analysis. The taller and late maturing mutants WP-30-1 and WP-30-5 were grouped together with IWP-control. WP-6-3, a mutant with high rice length after cooking and linear elongation ratio formed a separate cluster. The remaining mutants were grouped under the third cluster (Fig 3).

### Principal components analysis

Six principal components (PC1 to PC6) were extracted with Eigen values above one (Table 4, S5 Table). Together, these six components explained about 82.8% of the variance where, PC1 explained 25.75%. In PC1, rice length after cooking, thousand grain weight and rice length before cooking were the major contributors of variance (S5 Table). The biplot drawn using PC1 and PC2 (Fig 3) shows the relationship between traits and genotypes.

### Molecular analysis with SSR markers

The SSR marker panel consisting of 53 SSR markers showed the mutation rate in the mutants. Dissimilarity analysis showed the effect of mutagenesis at different SSR loci (S6 Table; S1 Fig). Totally, 71 alleles were recorded for the SSR markers screened. Sixteen markers were polymorphic in which 14 markers showed two alleles and two markers showed three alleles.

The dissimilarity values were generally low between the mutants and the IWP-control (S6 Table). The tall and late maturing mutants WP-30-1 and WP-30-5 were clustered in close

**Table 3. Mean performance of the M$_6$ mutants and IWP-control.**

| S. No. | Lines | Mill (%) | HRR (%) | LBC (mm) | BBC (mm) | L/B | LAC (mm) | BAC (mm) | LER | BER | ASV | GC (mm) | Amylose (%) |
|---|---|---|---|---|---|---|---|---|---|---|---|---|---|
| 1 | WP 5–1 | 68.7 | 61.1 | 5.00 | 2.00 | 2.50 | 7.85 | 2.70 | 1.57 | 1.35 | 3.0 | 84.8 | 33.5 |
| 2 | WP 5–4 | 65.7 | 57.7 | 4.95 | 2.00 | 2.48 | 7.80 | 2.90 | 1.58 | 1.45 | 3.0 | 100.0 | 22.3 |
| 3 | WP 6–3 | 64.6 | 54.2 | 5.05 | 1.95 | 2.59 | 8.05 | 2.80 | 1.59 | 1.44 | 4.0 | 57.7 | 16.5 |
| 4 | WP 6–4 | 68.6 | 60.4 | 5.10 | 2.10 | 2.43 | 7.50 | 2.70 | 1.47 | 1.29 | 3.0 | 61.5 | 34.8 |
| 5 | WP 6–5 | 66.6 | 55.6 | 5.20 | 2.05 | 2.54 | 7.80 | 2.70 | 1.50 | 1.32 | 3.0 | 100.0 | 36.6 |
| 6 | WP 15–1 | 65.1 | 58.0 | 5.00 | 1.95 | 2.57 | 7.35 | 2.50 | 1.47 | 1.28 | 3.0 | 87.2 | 26.2 |
| 7 | WP 15–5 | 67.1 | 61.5 | 5.15 | 2.00 | 2.58 | 7.50 | 2.75 | 1.46 | 1.38 | 3.0 | 80.9 | 23.0 |
| 8 | WP 16–1 | 68.0 | 62.1 | 5.05 | 2.00 | 2.53 | 7.30 | 2.70 | 1.45 | 1.35 | 3.0 | 65.4 | 32.7 |
| 9 | WP 16–2 | 67.5 | 61.2 | 5.00 | 2.00 | 2.50 | 7.15 | 2.60 | 1.43 | 1.30 | 3.0 | 100.0 | 37.2 |
| 10 | WP 16–3 | 68.4 | 56.2 | 4.85 | 2.00 | 2.43 | 7.35 | 2.75 | 1.52 | 1.38 | 3.0 | 78.9 | 35.6 |
| 11 | WP 16–4 | 68.4 | 60.3 | 4.90 | 1.95 | 2.51 | 7.55 | 2.50 | 1.54 | 1.28 | 3.0 | 100.0 | 29.1 |
| 12 | WP 16–5 | 68.5 | 57.4 | 5.20 | 2.00 | 2.60 | 7.40 | 2.60 | 1.42 | 1.30 | 3.0 | 74.7 | 21.8 |
| 13 | WP 22–1 | 67.5 | 57.6 | 5.05 | 2.00 | 2.53 | 7.25 | 2.40 | 1.44 | 1.20 | 3.0 | 100.0 | 30.0 |
| 14 | WP 22–2 | 66.7 | 57.6 | 4.95 | 1.85 | 2.68 | 7.20 | 2.50 | 1.45 | 1.35 | 4.0 | 100.0 | 25.8 |
| 15 | WP 22–3 | 66.7 | 58.7 | 5.20 | 1.95 | 2.67 | 7.55 | 2.60 | 1.45 | 1.33 | 3.0 | 88.1 | 34.5 |
| 16 | WP 22–5 | 68.2 | 60.9 | 4.85 | 1.80 | 2.70 | 7.30 | 2.40 | 1.51 | 1.33 | 3.0 | 100.0 | 29.3 |
| 17 | WP 23–3 | 67.2 | 57.2 | 5.00 | 2.05 | 2.44 | 7.35 | 2.80 | 1.47 | 1.37 | 3.0 | 75.3 | 13.2 |
| 18 | WP 23–4 | 68.7 | 57.7 | 4.95 | 2.00 | 2.48 | 7.40 | 2.70 | 1.50 | 1.35 | 3.0 | 100.0 | 31.2 |
| 19 | WP 30–1 | 62.5 | 50.4 | 5.45 | 2.05 | 2.66 | 8.40 | 2.65 | 1.54 | 1.29 | 3.0 | 87.9 | 15.0 |
| 20 | WP 30–5 | 65.3 | 58.6 | 5.50 | 2.00 | 2.75 | 8.15 | 2.45 | 1.48 | 1.23 | 3.0 | 75.1 | 31.5 |
| 21 | IWP Cont | 61.7 | 53.1 | 5.45 | 2.00 | 2.73 | 8.10 | 2.75 | 1.49 | 1.38 | 3.0 | 67.6 | 19.3 |
| | Grand Mean | 66.7 | 57.95 | 5.09 | 1.99 | 2.56 | 7.59 | 2.64 | 1.49 | 1.33 | 3.00 | 85.00 | 27.10 |
| Range | | 61.70 to 68.70 | 50.35 to 62.05 | 4.85 to 5.50 | 1.80 to 2.10 | 2.43 to 2.75 | 7.15 to 8.40 | 2.40 to 2.90 | 1.42 to 1.59 | 1.20 to 1.45 | 3.00 to 4.00 | 57.7 to 100.00 | 19.3 to 37.2 |
| PCV (%) | | 2.93 | 5.10 | 3.62 | 2.97 | 3.60 | 4.62 | 5.08 | 3.36 | 3.76 | | 17.01 | 26.79 |
| GCV (%) | | 3.01 | 5.15 | 3.78 | 3.73 | 4.19 | 4.68 | 5.49 | 3.36 | 5.04 | | 16.9 | 26.8 |
| Heritability (%) | | 94.83 | 98.33 | 91.89 | 63.64 | 73.91 | 97.62 | 85.71 | 100.00 | 55.56 | | 99.0 | 99.7 |
| Genetic advance | | 3.92 | 6.04 | 0.36 | 0.10 | 0.16 | 0.71 | 0.26 | 0.10 | 0.08 | | 29.5 | 15.2 |
| Genetic advance (as per cent of mean) | | 5.87 | 10.42 | 7.15 | 4.89 | 6.38 | 9.41 | 9.69 | 6.91 | 5.77 | | 34.7 | 55.0 |

(Mill-milling per cent; HRR-head rice recovery; LBC-dehusked kernel length before cooking; BBC-dehusked kernel breadth before cooking; L/B-length to breadth ratio; LAC-rice length after cooking; BAC-rice breadth after cooking; LER-linear elongation ratio; BER-breadth wise elongation ratio, ASV-alkali spreading value; GC-gel consistency).

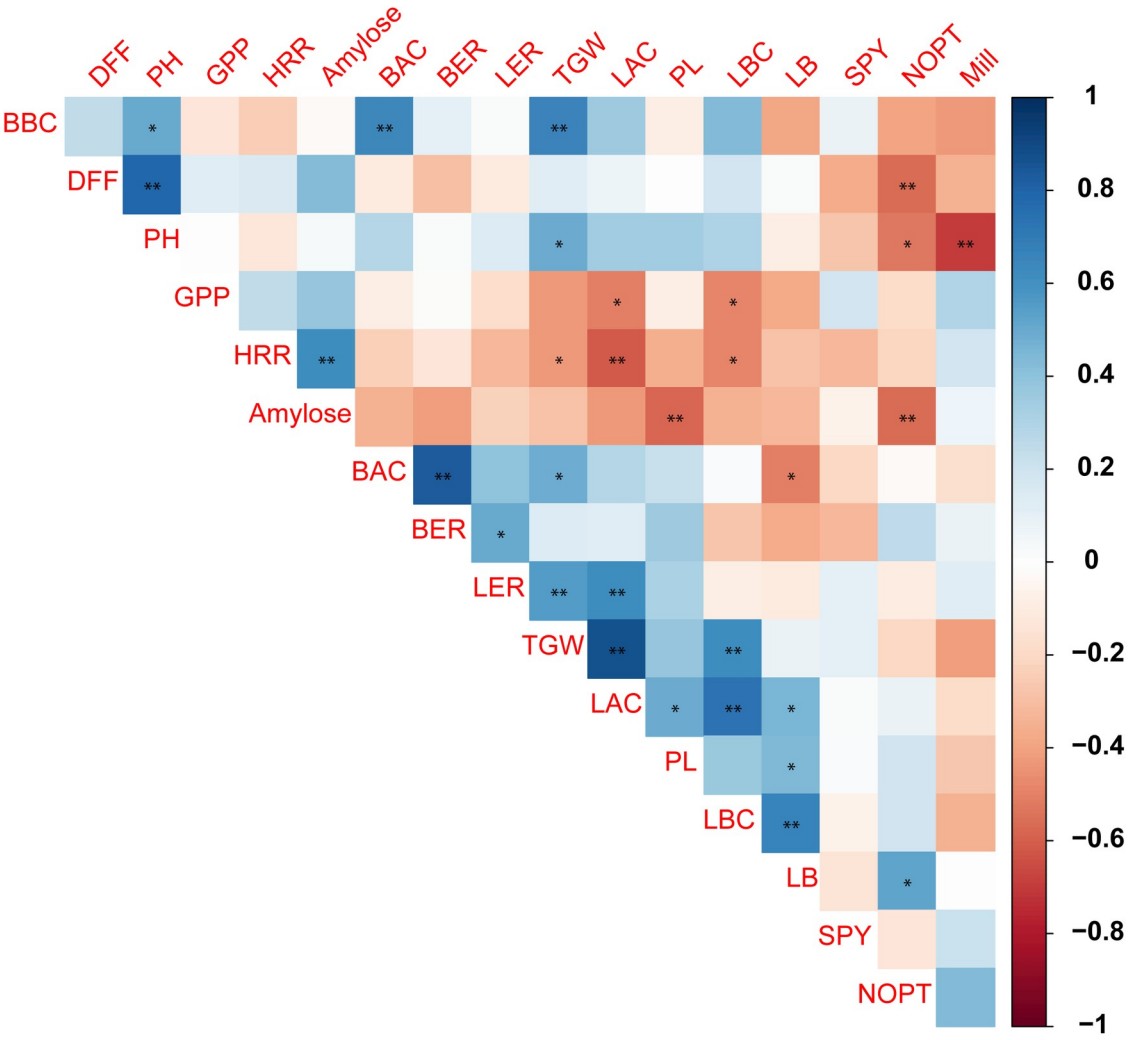

**Fig 2. Genotypic correlation between morphological traits.** The plant height has high positive correlation with days to fifty per cent flowering and high negative correlation with milling per cent. **-significant at 1% and *-5% level of significance.

affinity with IWP-control showing low dissimilarity (Fig 4). The genotypes WP-16-3 with WP-16-4 and WP-22-1 with WP-22-2 had low dissimilarity of 0.038. Maximum variability was observed between IWP-control and WP 23–3 with dissimilarity value of 0.45 (S6 Table).

## Responsiveness of mutant rice to external GA3

Based on overall morphological performance, WP-22-2 was selected as a superior mutant and was used in further characterisation studies. Morphological changes was observed in this mutant as a result of external GA$_3$ application. Significant increase in seedling height of WP-22-2 was contributed by the increase in 2$^{nd}$ leaf length (14.9 cm ± 0.99 S.E. in WP-22-2(GA$_3$); Fig 5; S7 and S8 Tables).

## Scanning electron microscopy

The SEM images showed cell patterning differences in the regions of internodes between IWP and a semi-dwarf and an early-maturing mutant. Large cell size and reduced number of cells per unit area was observed in the studied dwarf mutant (S2 Fig).

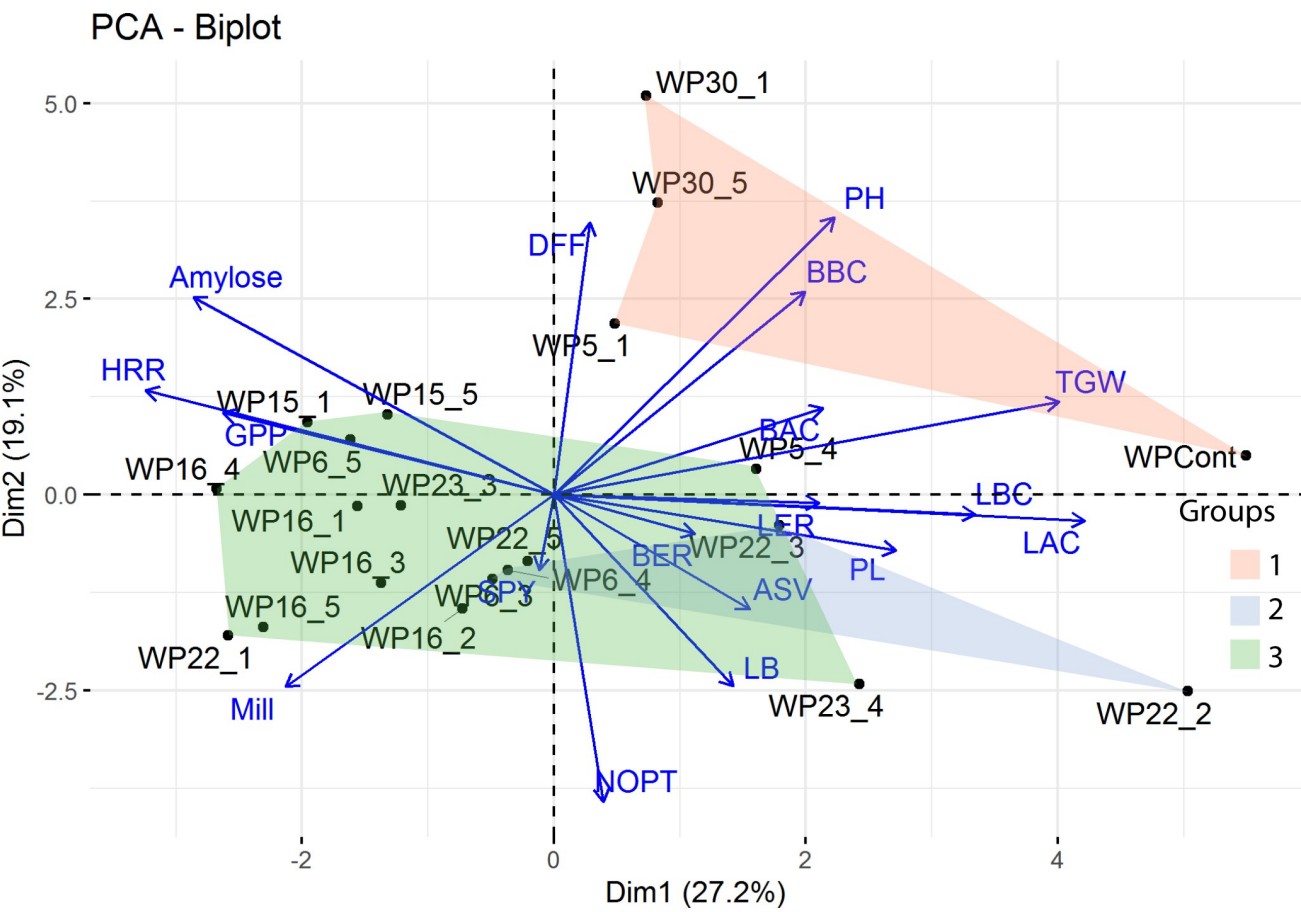

**Fig 3. PCA and hierarchical cluster analysis of genotypes based on morphological data.** The biplot was drawn using the first two principal components and overlaid with the three clusters produced by hierarchical clustering. The group 1 has tall and late-maturing genotypes; Group 2 has semi-dwarf mutants with similar panicle length and alkali spreading values; Group 3 has semi-dwarf and early maturing mutants with high grains per panicle and milling per cent. *(DFF-days to fifty per cent flowering; PH-plant height; BBC- dehusked kernel breadth before cooking; BAC-rice breadth after cooking; TGW-thousand grain weight; LBC-dehusked kernel length before cooking; LAC-rice length after cooking; LER-linear elongation ratio; PL-panicle length; BER-breadthwise elongation ratio; ASV-alkali spreading value; LB-length breadth ratio; NOPT-number of productive tillers; SPY-single plant yield; Mill-milling per cent; GPP-grains per panicle; HRR-head rice recovery).*

## Quantitative real time-PCR

The effect of external $GA_3$ at molecular level was compared between IWP and WP-22-2 at four time points (Fig 6; S9 and S10 Tables). Of the six genes compared, four genes (KOL4, KO2, MAX2 and BRD2) showed significant differences in expression levels between IWP and WP-22-2.

Downregulation of *SD1* gene (*GA20Ox2*), a key regulator of gibberellin pathway of rice was observed in both IWP and WP-22-2. The relative expression levels of OsKOL4 in IWP gradually decreased from 0 hr to 24 hrs after the $GA_3$ application. In mutant WP-22-2, the expression levels reduced from control (0 h); however, remained higher than IWP. Significant differences in expression levels of KO2 and MAX2 genes in IWP and WP-22-2 were witnessed. In IWP, the expression levels of KO2 gene remained higher than the WP-22-2. Gradual increase in expression of MAX2 gene was observed in WP-22-2. The BRD2 gene (brassinosteroid deficient) contrasting pattern of expression was observed in IWP (upregulation) and WP-22-2 (downregulation).

**Table 4. Principal components and per cent of variance explained.**

| Components | Eigenvalue | Variance (%) | Cumulative Variance (%) |
|---|---|---|---|
| PC1 | 4.89 | 25.75 | 25.75 |
| PC2 | 3.45 | 18.15 | 43.90 |
| PC3 | 2.70 | 14.20 | 58.10 |
| PC4 | 1.89 | 9.93 | 68.03 |
| PC5 | 1.64 | 8.62 | 76.65 |
| PC6 | 1.17 | 6.14 | 82.79 |
| PC7 | 0.98 | 5.18 | 87.97 |
| PC8 | 0.68 | 3.60 | 91.58 |
| PC9 | 0.56 | 2.96 | 94.53 |
| PC10 | 0.38 | 1.99 | 96.52 |
| PC11 | 0.29 | 1.53 | 98.06 |
| PC12 | 0.20 | 1.04 | 99.10 |
| PC13 | 0.09 | 0.48 | 99.58 |
| PC14 | 0.06 | 0.30 | 99.88 |
| PC15 | 0.02 | 0.10 | 99.98 |
| PC16 | 0.00 | 0.02 | 100.00 |
| PC17 | 0.00 | 0.00 | 100.00 |
| PC18 | 0.00 | 0.00 | 100.00 |
| PC19 | 0.00 | 0.00 | 100.00 |

## Targeted sequencing of OsGA20Ox2 gene

Whole genome assembly of IWP and WP-22-2 has indicated mutations in the exon 1 and intron 1 regions of *OsGA20Ox2* gene (preliminary analysis of whole genome assembly; results unpublished). Sanger sequencing of this gene revealed 356 bp deletion in WP-22-2 as against IWP (Fig 7). The bases 296 to 652 in exon1 and intron has been lost in WP-22-2.

## Discussion

The 'green revolution' gene in rice—the mutant gene of GA20ox2 (*sd1*)—has resulted in improved source-sink relationship in rice cultivars, which dramatically improved rice grain yield [3,8]. However, use of this gene as the single source of semi-dwarfism in rice cultivars has resulted in loss of diversity. Hence, identification of a new allele as a source of semi-dwarfism has long been researched and reported [8,17,38].

In popular rice cultivars, mutation breeding has been routinely used to improve the plant architecture. Since the mutation events are random throughout the genome, chances for identifying superior and novel alleles remain higher.

We employed gamma rays to develop semi-dwarf and early maturing mutants of the popular south Indian rice variety Improved White Ponni (IWP). The overall agronomic performances of the IWP mutants were better than the IWP-control as studied in the $M_6$ generation. The mean performance of IWP mutants indicates a significant reduction in plant height and days to flowering (Fig 1). Apart from these two, IWP mutants showed increased yield than IWP-control. A yield increase of up to 18.65 g (45.73%) was recorded in mutant WP-15-5. Seven mutants *viz.*, WP-15-1, WP-15-5, WP-16-1, WP-16-2, WP-16-4, WP-22-2 and WP-22-3 have recorded yield increase above 20% than IWP-control. The correlation of single plant yield with plant height and days to 50% flowering were negative. This clearly shows that the reduced plant height in the mutants improved the source-sink relationship.

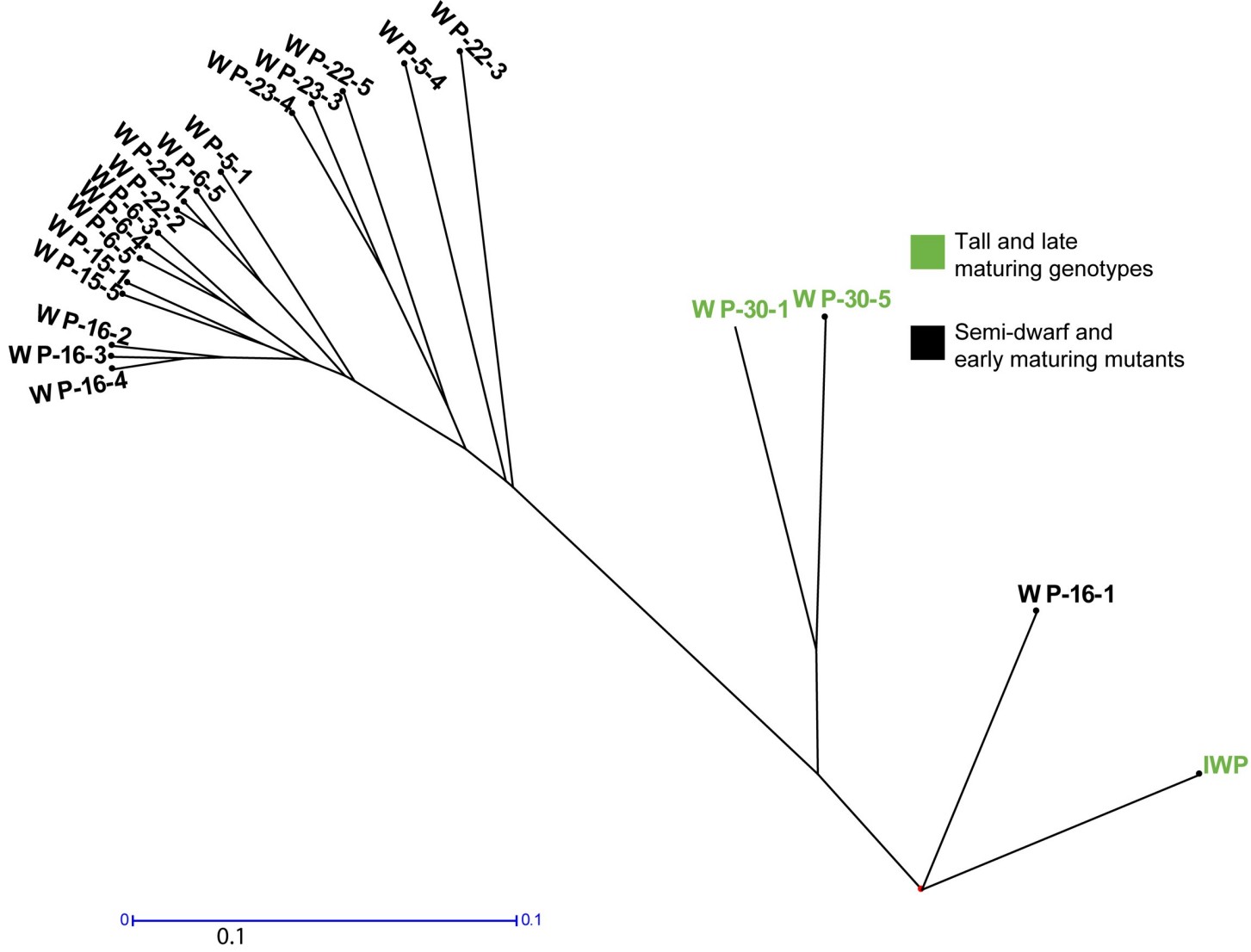

**Fig 4. Variability analysis using SSR marker data.** Genetic clustering with SSR marker data separated the semi-dwarf and early maturing mutants from the tall and late maturing genotypes. Cont (IWP), WP-30-5 and WP-30-1 are tall and late-maturing genotypes and other genotypes in the clusters are semi-dwarf and early maturing mutants. The scale bar indicates genetic distance of 0.1.

Although an increase in yield of cultivars is desired, maintaining the grain quality is highly important in a commercial perspective, especially with the highly preferred varieties like Improved White Ponni. All the mutants evaluated here outperformed IWP-control in milling per cent. Mutants WP-5-1, WP-6-4, WP-15-5, WP-16-1, WP-16-2, WP-16-4, WP-22-3, WP-22-5 and WP- 30–5 recorded more than 10% increase in head rice recovery. Amylose content had positively contributed to the increased head rice recovery in mutants (r = 0.61; $P = <0.01$) which was similar to the earlier reports [39,40].

However, dehusked kernel length was reduced in semi-dwarf and early maturing mutants. Non-significant positive correlation observed between plant height and days to flowering with dehusked kernel length before and after cooking suggests a relationship between these traits (S4 Table). But the breadth remained unaltered or reduced among the IWP mutants. Hence, there was no major changes in the L/B ratio and fifteen mutants and IWP-control with L/B ratio above 2.50 can be grouped under medium grain rice. The fine slender grain trait

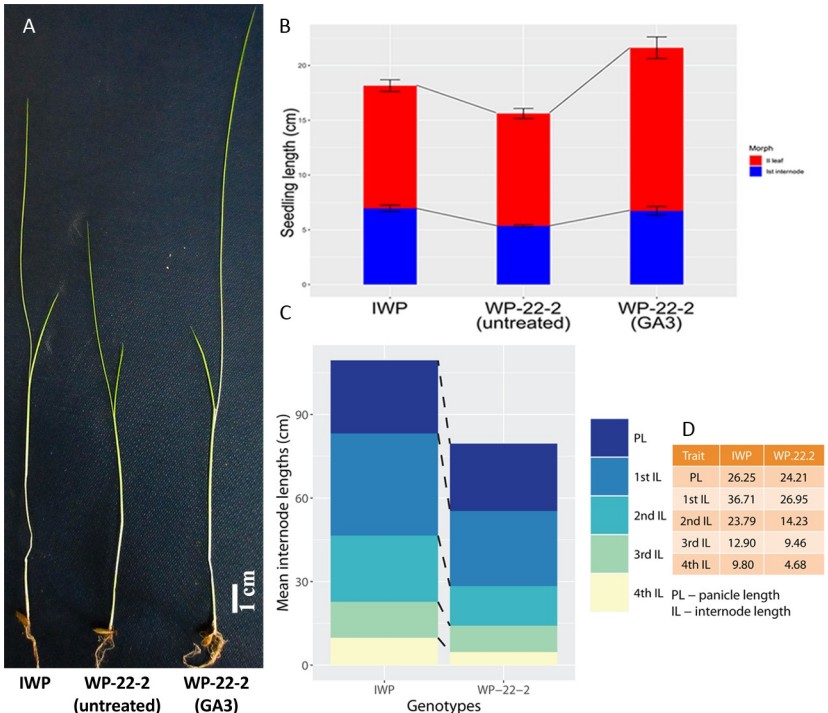

**Fig 5. Responsiveness of WP-22-2 mutant to external-GA₃.** 50 µM GA₃ was sprayed on 10 day old seedlings of WP-22-2 which completely reverted the plant height similar to Improved White Ponni. A) Comparison of IWP, WP-22-2 (untreated) and WP-22-2 (GA₃ treated) seedlings after fourteen days from sowing; scale bar– 1 cm; B) Stacked barplot representing the seedling growth between IWP, WP-22-2 (untreated) and WP-22-2 (GA3 treated). While the primary node lengths were similar, second leaf length showed much variation (red indicates II leaf length and blue indicates the 1$^{st}$ internode). The error bars indicate standard error of the mean for three independent experiments (N = 3); C) Stacked barplot showing the panicle length and first four internode lengths of IWP and WP-22-2 after maturity. Uniform reduction in the length was observed in four internodes of WP-22-2 responsible for semi-dwarfism; D) Table showing the internode lengths of IWP and WP-22-2; unit–cm.

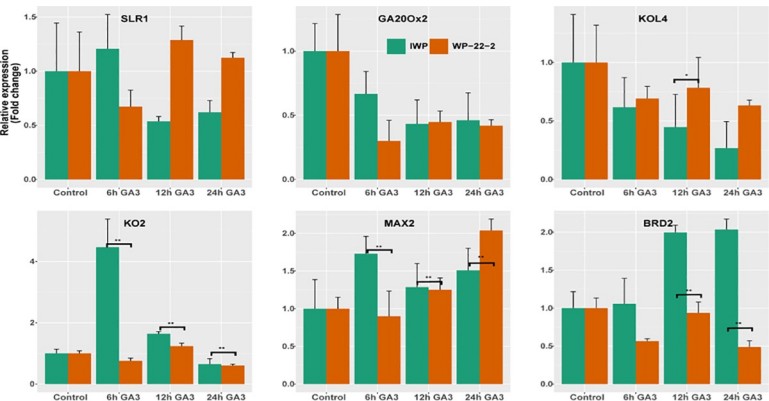

**Fig 6. Graph of gene expression.** The relative expression levels of six genes controlling plant height in rice were studied. Clear variations in expression levels are visible in ent-kaurene oxidase 2 (KO2), MAX2 and OsBRD2 genes. Interestingly, both the IWP and WP-22-2 showed reduced expression levels in sd1 gene (GA20Ox2). Error bars indicate standard error (N = 3).

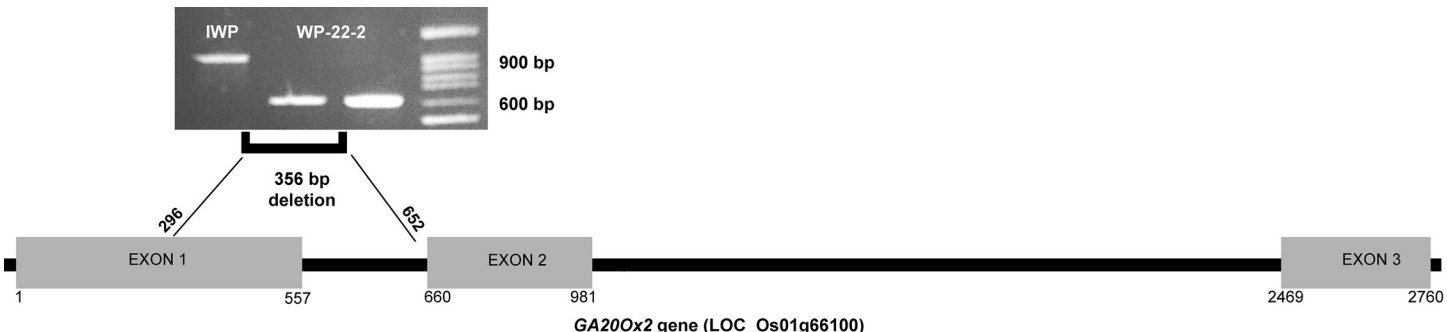

**Fig 7. Deletion in *GA20Ox2* gene of WP-22-2 mutant.** Deletion in exon 1-intron 1 regions was identified in targeted gene sequencing of *GA20Ox2*. Electrophoresis image shows the amplicon differences between IWP and WP-22-2. Gene diagram shows the relative positions of three exons and introns of the genes; and the position of deletions in the gene (horizontal black bar indicates introns and grey boxes indicate exons).

IWP-control was maintained among the mutants (Fig 1C). Rice varieties with more linear expansion and less breadth wise ratio have high preference. These two traits among the mutants suggested less change in rice length after cooking.

The other cooking qualities of rice, gelatinization temperature of rice measured by alkali spreading value (ASV) and gel consistency was similar for IWP and the mutants. Intermediate gelatinization temperature (ASV score of '3') and soft gel consistency has high preference in many rice growing countries [41]. Further, many mutants had intermediate amylose similar to the IWP (Table 3). Mutant WP-22-2 shows balanced morphological improvements such as semi-dwarfism, earliness, high milling and head rice recovery, high L/B ratio and especially higher yield than IWP. Hence, this mutant was mainly selected as a superior mutant. SSR markers are powerful tools to study the polymorphism created by mutagenesis [42–45]. In addition, they are useful to identify true mutants from outcrosses or mixtures [46,47]. The strategy to select 53 SSR markers (S2 Table) with known associations with plant height and days to maturity QTLs [48–66] was highly useful since they have clearly differentiated between the semi-dwarf and early maturing genotypes from wild-types (S1 Fig). Clustering based on morphological data (Fig 3) and molecular marker data (Fig 4) have clearly separated the IWP-control and tall, late maturing mutants (WP-5-1, WP-30-1 and WP-30-5) in same clusters. Out of the 53 SSR markers used, only two markers (RM302 and RM310) showed maximum variability *i.e.* three alleles. This is expected since the mutant population is derived from the single parent *viz.*, IWP [46,47]. Even the highest genetic distance of 0.45 (IWP with WP-23-3) was very low (given the range of dissimilarity: 0 to ∞), an indication that induced mutations cannot create drastic variations. With this, WP-23-3 can be considered as the genotype with more mutations for the SSR loci tested. Similar results with SSR markers were observed for other rice mutants created with *N*-Nitroso-*N*-methylurea [45], somaclonal mutants [67] and ion beam radiation [68].

Intercalary meristem cell division and elongation are the major causes for internodal elongation rice and flaws in these processes severely affect the plant height. Many studies in dwarf mutants suggest defects in gibberellic acid pathway that reduce the cell division [21]. $GA_3$ treatment can restore the plant height in mutants, a characteristic feature in GA deficient mutants [69]. This same feature was observed in $GA_3$ treated WP-22-2 suggesting its $GA_3$ deficiency. At the two leaf stage of the plants with which the experiment was performed, the 2nd leaf was highly responsive to the $GA_3$ than the 1st internode (Fig 5A and 5C). Thus it was the major contributor of plant height in IWP. The scanning electronic microscopic images show reduced number of cells per unit area in the mutants. This explains the reduction in internode lengths in rice mutants. These experiments hinted that there is a deficient gibberellin pathway in WP-22-2 causing semi-dwarfism.

To further study the molecular level changes during gibberellin treatment relative expression of six genes was compared using qRT-PCR (Fig 6). Overall, the expression changes between IWP and WP-22-2 was indicative of a deficient gibberellin pathway in WP-22-2. The *SLENDER1* (SLR-1) gene of rice (DELLA protein SLR1-like) interacts with GA-GID1 complex to act as a GA signalling repressor [8] and overexpression induces dwarf phenotype. Gradually decreasing time-course expression levels in IWP and increasing expression levels in WP-22-2 indicated mutations. Comparison of *GA20Ox2* gene (GA20oxidase), the major semi-dwarfing gene utilised in breeding of rice has indicated an interesting phenomenon. In plants, the GA20oxidase converts GA intermediates into bioactive forms [7]; hence loss of function may cause dwarfism in rice plants. But, reduced expression in both IWP (wild-type) and WP-22-2 indicated possible mutations in other dwarfing regions such as the alternate semi-dwarf 1 [70] and Slr-d6 [71]. Such mutants were reported to be responsive to the $GA_3$ hormone to a limited extent [71] or a key regulator in the Brassinosteroid pathway of rice [70]. But, the $GA_3$ responsiveness of the mutant studied here is prominent, a characteristic feature of mutations in the $GA_3$ pathway of rice [8]. It further necessitates the requirement of genome-wide characterisation of the mutant to identify other alleles causing semi-dwarfism.

Significant differences in expression levels were observed for four genes: OsKOL4, OsKO2, MAX2 and OsBRD2. Of these, *ent*-kaurene oxidases (OsKO2) and *ent*-kaurene oxidase like proteins (OsKOL4) regulate the gibberellic acid pathway by converting the intermediary *ent*-kaurene to $GA_{12}$ [6]. Inhibited activity and/or expression of the *ent*-kaurene oxidase like proteins result in dwarfism in rice [38]. This is clearly witnesses with OsKO2: in IWP, the gene is over expressed while in WP-22-2 it is down regulated, suggesting mutations in the gene. Similar expression changes were witnessed for Fbox LRR/MAX2 protein (an orthologous gene of *Arabidopsis* MAX2/ORE9) which is known to control apical dominance. Mutations in these regions cause dwarfism and high tillering in rice [72]. The increased expression in both the genotypes is an indicator of $GA_3$ responsiveness of the mutant.

A more prominent pattern of expression changes can be seen in the OsBRD2 gene (delta 24-sterol reductase) which is a part of the Brassinosteroid pathway and reduction in activity results in dwarfism. This reduced activity (downregulation) is observed in WP-22-2 indicating mutations in the gene. Hence, the semi-dwarfism in WP-22-2 can be attributed to deficiencies in the two independent gibberellin and brassinosteroid pathways. However, it is necessary to do genome wide characterisation of this mutant. This mutation is significant as it is useful to reduce the dependency on *GA20Ox2* as a single dwarfing gene. Further, this complex control will reduce the genetic bottlenecking effect in rice cultivars. of plant height with high yield and fine slender grain quality in WP-22-2 could reduce the dependency on the single gene, the *GA20Ox2* for semi-dwarfism in other high grain quality rice cultivars.

Our preliminary whole genome resequencing analysis (results unpublished) in IWP and WP-22-2 has shown 356 bp deletion in *GA20Ox2* (LOC_01g66100) gene (Fig 7) which could be responsible for the semi-dwarfism in WP-22-2 mutant. Mutations in *GA20Ox2* gene (*sd1* alleles) are generally deficient in GA metabolism and cause semi-dwarfism in rice plants [8]. Although the large deletion could be associated with the semi-dwarfism in WP-22-2, expression of this gene was not much variable between IWP and WP-22-2. Epistatic interactions of the other genes studied in qRT PCR could explain these differences. However, further investigation with the whole genome information would clearly explain the behaviour.

## Conclusion

The study summarizes the positive effect of gamma rays on the plant architecture of Improved White Ponni. We identified several mutants with semi-dwarfism and earliness of which, WP-

22-2, WP-15-5, WP-16-1, WP-16-1 and WP-15-1 were superior for agronomic traits and commercially important grain quality traits. Molecular level mutations were confirmed with SSR markers which produced clusters similar to morphological clustering. Based on overall performance, we propose WP-22-2 mutant in place of IWP-control with increased tolerance to lodging and with high yield. Even though the semi-dwarf mutant WP-22-2 was gibberellin responsive, a possible epistatic control (between the genes of gibberellin and brassinosteroid pathway) rather than an effect of a single gene was witnessed. This may preserve the valuable genetic diversity by reducing the dependence on *OsGA20Ox2* gene. However, a genome wide characterisation study is required to further validate this data.

## Supporting information

**S1 Fig. Electrophoresis images of polymorphic SSR markers used in the study.**
(PDF)

**S2 Fig. Scanning electron microscopy of Improved White Ponni and a dwarf mutant.**
(PDF)

**S1 Table. Categories of rice based on amylose content (%).**
(PDF)

**S2 Table. List of SSR markers used in this study.**
(PDF)

**S3 Table. Category of IWP and mutants based on amylose content.**
(PDF)

**S4 Table. Genotypic correlation between traits and their significance.**
(PDF)

**S5 Table. Contribution of traits to the variances of principal components.**
(PDF)

**S6 Table. Dissimilarity matrix constructed based on microsatellite marker data.**
(PDF)

**S7 Table. Mean internode and second leaf lengths of IWP, WP-22-2 (untreated) and WP-22-2 (GA3 treated).**
(PDF)

**S8 Table. Student's t-test for variances of $1^{st}$ internode and $2^{nd}$ leaf lengths.**
(PDF)

**S9 Table. The raw $C_T$ values and the calculated $\Delta\Delta C_T$ values observed for Improved White Ponni.**
(PDF)

**S10 Table. The raw $C_T$ values and the calculated $\Delta\Delta C_T$ values observed for WP-22-2.**
(PDF)

## Acknowledgments

We thank Dr.Roshan Kumar Singh, and Dr. Manoj Prasad of National Institute of Plant Genome Research, New Delhi, India for providing lab space and the precious help rendered during qRT-PCR analysis. We gratefully acknowledge Dr. Ganesh Ram, Professor, Tamil Nadu Agricultural University for critical reading of this manuscript.

## Author Contributions

**Conceptualization:** M. Arumugam Pillai.

**Data curation:** Andrew-Peter-Leon M. T., S. Ramchander, Kumar K. K., Mehanathan Muthamilarasan.

**Formal analysis:** Andrew-Peter-Leon M. T., S. Ramchander, Kumar K. K., Mehanathan Muthamilarasan.

**Investigation:** M. Arumugam Pillai.

**Methodology:** Kumar K. K., Mehanathan Muthamilarasan, M. Arumugam Pillai.

**Resources:** Mehanathan Muthamilarasan, M. Arumugam Pillai.

**Software:** Andrew-Peter-Leon M. T.

**Supervision:** M. Arumugam Pillai.

**Validation:** Mehanathan Muthamilarasan, M. Arumugam Pillai.

**Writing – original draft:** Andrew-Peter-Leon M. T.

**Writing – review & editing:** Andrew-Peter-Leon M. T., M. Arumugam Pillai.

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
