## [Decision Letter · Decision Letter 0]

28 Oct 2020

PONE-D-20-23412

Gamma ray induced mutants of rice 'Improved White Ponni' - with semi-dwarfism, high yield and early maturity

PLOS ONE

Dear Dr. Pillai,

Thank you for submitting your manuscript to PLOS ONE. After careful consideration, we feel that it has merit but does not fully meet PLOS ONE’s publication criteria as it currently stands. Therefore, we invite you to submit a revised version of the manuscript that addresses the points raised during the review process.

Address each point raised by the reviewers for additional information, clarifications, and general editing.  Pay particular attention to the following -

Reviewer #1's points on the strength of the evidence that these lines may not have been derived from the gamma ray treatment.  For example, how many SSR markers are typically used to confidently support mutant identification and how this relates to using 53 SSR markers? It may be necessary to adjust the claims. Reviewer #3's point to rewrite the M&M to provide a "complete description of experimental design, rice ecosystem evaluated, year of evaluation etc." 

We look forward to receiving your revised manuscript.

Kind regards,

Randall P. Niedz

Academic Editor

PLOS ONE

Journal Requirements:

2. Please upload a copy of Figure 6 and Figure 8-a, to which you refer in your text on page 18. If the figure is no longer to be included as part of the submission please remove all reference to it within the text.

3. Please include a caption for figure Figure 8-a.

Reviewers' comments:

Reviewer's Responses to Questions

**Comments to the Author**

1. Is the manuscript technically sound, and do the data support the conclusions?

Reviewer #1: Partly

Reviewer #2: Yes

Reviewer #3: Partly

2. Has the statistical analysis been performed appropriately and rigorously? 

Reviewer #1: Yes

Reviewer #2: Yes

Reviewer #3: I Don't Know

3. Have the authors made all data underlying the findings in their manuscript fully available?

Reviewer #1: Yes

Reviewer #2: Yes

Reviewer #3: Yes

4. Is the manuscript presented in an intelligible fashion and written in standard English?

Reviewer #1: Yes

Reviewer #2: Yes

Reviewer #3: No

5. Review Comments to the Author

Reviewer #1: The manuscript describes 20 semi-dwarf and early-maturing rice mutants induced by gamma ray. The differences between the mutants and the wild type ‘Improved White Ponni’ were morphologically and physiologically studied. The authors also used 53 pairs of SSR markers to genetically analyze the mutants. Especially, one particular mutant was selected to check its response to GA and the expressions of GA-related genes.

This is an interesting work about germplasm improvement in rice, however, there are still some doubts for the candidate mutants.

1\\ Were these mutants real? The author used only 53 pairs of SSR markers to do genetic identification. That’s not adequate to support its identity as a real mutant. Higher density of markers will be needed. Based on the polymorphism results in Table S6, I consider that some of these lines maybe not derived from gamma ray induced, for instance, the dissimilarity between WP 23-3 and WT is 0.453 and beyond the standard of reliability for real mutants. The polymorphism value of credibility mutant is less than 1% based on some published resequencing data of rice mutants.

2\\What is the causal mutant gene in the mutant? Although the response to GA and the expressions of the known GA-related genes were studied, the causal mutated genes in the mutant was not confirmed. In another way, the mutation mechanism is still unclear. It will be more valuable if the gene mapping of the mutants is done.

Reviewer #2: Line 1 : Gamma ray to be corrected as Gamma rays and in other places

Line 140: ten days old seedling and in Line 147 for field study, fourteen days old seedling for qrtPCR, why the same day old seedling not used for both GA3 study

This study paddy grain length and breadth or kernel length and breadth are measured, correction can be made accordingly.

Line 174: (Table 2 and 3), 3 to be removed

Table 1: Primer ID or gene ID; Sequence (5 �3) can be write as Primer Sequence (5 �3)

Table 2 and 3: PCV and PCV mention in (%), G.M. as Grand mean

Line 249 to 251, the PC values and the eigen value mentioned in text and table is not matching

Line 274, mentioned as no significant difference in first intenode length between mutant and IWP, but in line 384 the increase in plant height due to second leaf length. That means there is no intermodal length difference between mtutant and IWP. Clarification needed

Reviewer #3: The research conducted has significance to rice breeding in terms of new variability created through mutation breeding. However, the manuscript needs to be revised thoroughly and my observations are:

1. Title is general and should highlight the work depicted in the manuscript.

2.The review of literature w.r.t to semidwarf mutants in rice to be mentioned in the introduction

3.The material and methods needs to be rewritten with complete description of experimental design, rice ecosystem evaluated, year of evaluation etc.

4.The checks for molecular screening of different semi-dwraf alleles, flowering that have been used to depict their pattern and comparison with mutants studied here to be mentioned , if any.

4.Results :As depicted in material and methods , the molecular markers are linked to plant height and flowering. However, diversity analysis of markers does not depict the causal linked markers in the mutants for plant height and flowering

5.Discussion: Line 419-427: Needs to be supported with linked SSR markers used.

6.Figures need to be provided with titles, Figure S1 :Only relevant gel pictures with significance in the study may be provided and their pattern to be highlighted, needs proper labelling of lanes.

6. PLOS authors have the option to publish the peer review history of their article (what does this mean?). If published, this will include your full peer review and any attached files.

Reviewer #1: No

Reviewer #2: **Yes: **P. Arunachalam, Assistant Professor, Plant Breeding and Genetics, Tamilnadu Agricultural University, Coimbatore

Reviewer #3: No

---

## [Author Response · Author response to Decision Letter 0]

20 Dec 2020

Reviewer 1

1) Were these mutants real? The author used only 53 pairs of SSR markers to do genetic identification. That’s no adequate to support its identity as a real mutant. Higher density of markers will be needed. 

• Thanks for the suggestion. Please refer to the Table S2, from which the reviewer can understand the panel of 53 SSR markers used in this study is generated with markers chosen from other QTL association studies. All these 53 SSR markers are proven to be closely linked with different QTLs governing the two traits of interest namely semi-dwarfism and earliness. As the chosen markers are proven to be associated with the traits, here we conclude this panel of 53 SSR markers are sufficient to assess the mutation rates in our M6 population of 20 families

2) Based on the polymorphism results in Table S6, I consider that some of these lines maybe not derived from gamma ray induced, for instance, the dissimilarity between WP-23-3 and WT is 0.453 and beyond the standard of reliability for real mutants. The polymorphism value of credibility mutant is less than 1% based on some published resequencing data of rice mutants.

• The dissimilarity score computed through SSR marker polymorphism is an indicator of variability created through mutagenesis. The dissimilarity range of 0.038 to 0.453 indicates a family specific polymorphism range as compared to wild-type with reference to the alleles under consideration. Whereas, the resequencing cut-off of 1% and below values referred by the reviewer is a genome-wide indicator which doesn’t have any significant implication with our approach. The high values of polymorphism arrived through our SSR analysis indicates significant variations induced by gamma-irradiation in the desired genomic regions. This has been suitably indicated in the discussion section 

3) What is the causal mutant gene in the mutant? Although the response to GA and the expressions of the known GA-related genes were studied, the causal mutated genes in the mutant were not confirmed. In another way, the mutation mechanism is still unclear. It will be more valuable if the gene mapping of the mutants is done.

• Thanks for the comment. The major objective of this study was to develop a mutant with semi-dwarfism and earliness in maturity without any compromise in yield and grain quality. It was largely successful after the identification of WP-22-2 as a superior mutant with all the above mentioned traits.

• The qRT-PCR was performed to compare the expression levels of six genes (with known association with plant height) between the mutant WP-22-2 and IWP. This has shown a general idea that there are significant differences in expression levels of these genes (S9 and S10 Tables; Fig 6). Hence, we conclude that there is an indication of mutations in these six genes.

• Our preliminary whole genome resequencing analysis (results unpublished) in IWP and WP-22-2 has shown 356 bp deletion in GA20Ox2 (LOC_01g66100) gene (Fig 7) which could be responsible for the semi-dwarfism in WP-22-2 mutant. This has now been included in the manuscript (lines 186 to 191).

Reviewer 2

1) Line 1: Gamma ray to be corrected as Gamma rays and in other places

• As per the suggestion, corrections have been made in appropriate places.

2) Line 140: ten days old seedling and in Line 147 for field study, fourteen days old seedling for qRTPCR, why the same day old seedling not used for both GA3 study

• In 14 days, the seedlings have reached the optimum growth: two leaf stage, at which gibberellin sensitivity studies were performed

• To visualize the morphological effects of the GA3 on seedlings, 10 days old seedlings were sprayed and allowed to grow for five more days (i.e. 15 days)

• Whereas, in qRT-PCR study, 14 days old seedling was taken in-line with the above experiment and the sample collection ended at the 15th day (0h to 24h with 6hrs interval)

3) Paddy grain length and breadth or kernel length and breadth are measure, correction can be made accordingly

• Thanks for the suggestion. We have changed the grain length and breadth as kernel length and breadth as per the suggestion

4) Line 174: (Table 2 and 3), 3 to be removed

• Thanks for the comments. We have made the correction as per suggestion. (Line no. 195)

5) Table 1: Primer ID or gene ID; Sequence (5 to 3) can be written as Primer Sequence (5’ to 3’)

• The corrections were made as per suggestion (Line 184).

6) Table 2 and 3: PCV and GCV mention in (%), GM as Grand Mean

• Appropriate changes have been made as per suggestions (Tables 2 and 3)

7) Line 249 to 251, the PC values and the eigen value mentioned in text and table is not matching

• We regret this mistake. The summary has been modified appropriately. (Line 275-281)

8) Line 274, mentioned as no significant difference in first internode length between mutant and IWP, but in line 384 the increase in plant height due to second leaf length. That means there is no internodal length difference between mutant and IWP. Clarification needed.

• Before GA3 treatment, the 2nd leaf length of IWP and WP-22-2 was similar (non-significant). After GA3 treatment however, the 1st internode length of WP-22-2 attained a similar 1st internode length of IWP and was non-significant.

• But the plant height of GA3 treated WP-22-2 has exceeded the IWP. This increased plant height observed was due to the higher 2nd leaf length than IWP.

• The sentences have been now modified for clarity. (lines 300 to 305).

Reviewer 3

1) Title is general and should highlight the work depicted in the manuscript

• The title has been changed to ‘Assessment of efficacy of mutagenesis of gamma-irradiation in plant height and days to maturity through expression analysis in rice‘

2) The review of literature w.r.t to semidwarf mutant in rice to be mentioned in the introduction

• We are thankful for the suggestion. The introduction has been modified with literatures semi-dwarfism in rice (lines 64 to 74).

3) The material and methods needs to be rewritten with complete description of experimental design, rice ecosystem evaluated, year of evaluation etc.

• Thanks for the suggestion. We have modified the materials and methods section with information about the development of mutants, year of experiment etc. (lines 89 to 109)

4) The checks for molecular screening of different semi-dwarf alleles, flowering that have been used to depict their pattern and comparison with mutants studied here to be mentioned, if any.

• The parent variety Improved White Ponni (a tall and medium to long duration cultivar) was used as the control in all the experiments; however, no other checks have been used to characterize the mutants. 

5) Results: As depicted in material and methods, the molecular markers are linked to plant height and flowering. However, diversity analysis of markers does not depict the causal linked markers in the mutants for plant height and flowering

• In this context we wish to inform that the mutant were screened from M2 generation by imparting selection. In M3 desirable mutants were checked phenotypically as well as by genotypic markers (Image 1) We got more variation at phenotypic as well as genotypic level and we are interested in isolating early and semidwarf mutant (since wild type is late, lodging and tall genotype). The similar banding pattern of the selected genotype in M3 (image 1) for RM302 and RM310 and M6 generation (image 2) indicating the homozygosity of this genotype in M6 generation. Some of the semidwarf mutant based on this pattern of selection is in advanced stage of testing and variety release

Image 1&2: Marker pattern of RM302 & RM310 in M3 generation of mutants

6) Discussion: Line 419-427: Needs to be supported with linked SSR markers used

• We would like to inform that the choice of SSR markers and the genes targeted in qRT-PCR were not related. Since SSR markers for rice are abundant and to select markers that can show good variability among the mutants, we have chosen some SSR markers that were reported to be linked with plant height and maturity. However, the SSR markers were only used to study the variability and no attempt was made to link them to the phenotypes observed in the mutants. 

• However, the qRT PCR was performed for some of the common genes that affect rice plant height.

• And so, the results of these experiments are discussed separately.

7) Figures need to be provided with titles. Figure S1: Only relevant gel pictures with significance in the study may be provided and their pattern to be highlighted, needs proper labeling of lanes.

• As per the suggestion, only the polymorphic marker images have been labeled and included (S1 Fig; p.no. 10 of supplementary information)

Additional comments:

1) Please ensure that your manuscript meets PLOS ONE’s style requirements, including those for file naming

2) Please upload a copy of Figure 6 and Figure 8-a, to which you refer in your text on page 18. If the figure is no longer to be included as a part of the submission please remove all reference to it within the text

• The typographical error Figure 8-a has been removed from the manuscript

3) Please include a caption for figure Figure8-a

• The typographical error Figure 8-a has been removed.

---

## [Editor Report · Decision Letter 1]

5 Jan 2021

Assessment of efficacy of mutagenesis of gamma-irradiation in plant height and days to maturity through expression analysis in rice

PONE-D-20-23412R1

Dear Dr. Pillai,

We’re pleased to inform you that your manuscript has been judged scientifically suitable for publication and will be formally accepted for publication once it meets all outstanding technical requirements.

Kind regards,

Randall P. Niedz

Academic Editor

PLOS ONE
---

## [Editor Report · Acceptance letter]

7 Jan 2021

PONE-D-20-23412R1 

Assessment of efficacy of mutagenesis of gamma-irradiation in plant height and days to maturity through expression analysis in rice 

Dear Dr. Pillai:

I'm pleased to inform you that your manuscript has been deemed suitable for publication in PLOS ONE. Congratulations! Your manuscript is now with our production department. 

Kind regards, 

on behalf of

Dr. Randall P. Niedz 

Academic Editor

PLOS ONE